# Development of a New Laparoscopic Detection System for Gastric Cancer Using Near-Infrared Light-Emitting Clips with Glass Phosphor

**DOI:** 10.3390/mi10020081

**Published:** 2019-01-24

**Authors:** Shunko A. Inada, Hayao Nakanishi, Masahiro Oda, Kensaku Mori, Akihiro Ito, Junichi Hasegawa, Kazunari Misawa, Shingo Fuchi

**Affiliations:** 1Department of Mechanical Science and Engineering, Faculty of Science and Technology, Hirosaki University, 3 Bunkyo-cho, Hirosaki 036-8561, Japan; 2Laboratory of Pathology and Clinical Research, Aichi Cancer Center Aichi Hospital, Okazaki 444-0011, Japan; hnakanis@aichi-cc.jp; 3Graduate School of Informatics, Nagoya University, Nagoya 464-8601, Japan; moda@mori.m.is.nagoya-u.ac.jp (M.O.); kensaku@is.nagoya-u.ac.jp (K.M.); 4Research Center for Medical Bigdata, National Institute of Informatics, Tokyo 101-8430, Japan; 5Graduate School of Pharmaceutical Sciences, Meijo University, Nagoya 468-0073, Japan; itou.pha@chubuh.johas.go.jp; 6Department of Media Engineering, School of Engineering, Chukyo University, Nagoya 466-8666, Japan; hasegawa@sist.chukyo-u.ac.jp; 7Department of Gastroenterological Surgery, Aichi Cancer Center Hospital, Nagoya 464-8681, Japan; misawakzn@aichi-cc.jp; 8Department of Electrical Engineering and Electronics, College of Science and Engineering, Aoyama Gakuin University, Kanagawa 252-5258, Japan; fuchi@ee.aoyama.ac.jp

**Keywords:** gastric cancer, laparoscopic surgery, fluorescent clip, near-infrared fluorescence imaging

## Abstract

Laparoscopic surgery is now a standard treatment for gastric cancer. Currently, the location of the gastric cancer is identified during laparoscopic surgery via the preoperative endoscopic injection of charcoal ink around the primary tumor; however, the wide spread of injected charcoal ink can make it difficult to accurately visualize the specific site of the tumor. To precisely identify the locations of gastric tumors, we developed a fluorescent detection system comprising clips with glass phosphor (Yb^3+^, Nd^3+^ doped to Bi_2_O_3_-B_2_O_3_-based glasses, size: 2 mm × 1 mm × 3 mm) fixed in the stomach and a laparoscopic fluorescent detection system for clip-derived near-infrared (NIR) light (976 nm). We conducted two ex vivo experiments to evaluate the performance of this fluorescent detection system in an extirpated pig stomach and a freshly resected human stomach and were able to successfully detect NIR fluorescence emitted from the clip in the stomach through the stomach wall by the irradiation of excitation light (λ: 808 nm). These results suggest that the proposed combined NIR light-emitting clip and laparoscopic fluorescent detection system could be very useful in clinical practice for accurately identifying the location of a primary gastric tumor during laparoscopic surgery.

## 1. Introduction

Gastric cancer is a major cause of disease-related death worldwide, causing the deaths of 723,000 people in 2012 [1], and it is the most common type of cancer in East Asia [2]. Among several options for the treatment of gastric cancer, such as surgery, radiotherapy and chemotherapy, surgical resection is the most reliable. Although open surgery has long been the main form of treatment for gastric cancer, laparoscopic surgery is now recognized as an alternative surgical modality for gastric cancers. Compared with open surgery, laparoscopic surgery is less invasive, results in good quality of life for the patient, and is almost equally effective [3,4,5].

In conventional laparoscopic surgery for gastric cancer, charcoal ink (activated carbon) or India ink is endoscopically injected around the primary gastric tumor to enable the visualization of the location of the tumor in the stomach intraperitoneally [6,7]. However, it is sometimes difficult to visualize the exact location of the primary tumor during the laparoscopic operation, as the injected charcoal ink diffusely spreads to areas distant from the tumor in the stomach, unlike in tumors of the colon. This charcoal ink diffusion may result in resection with a broad margin containing wide tumor-free tissue, which results in great inconvenience for the patient. To overcome this problem, we developed a new convenient method with which to visualize the exact location of a primary gastric tumor from outside the stomach.

In the present study, we focused on the near-infrared (NIR) wavelength around the 1000 nm band, which has good biological transmission activity [8], and developed a new fluorescent chip with glass phosphor and a laparoscopic fluorescent detection system to assist in the performance of safe and accurate laparoscopic removal of gastric cancer. Two ex vivo experiments performed in this pilot study demonstrated the successful use of this new detection system.

## 2. Materials and Methods

### 2.1. Development of Glass Phosphor

To obtain a biological transmission light with a wavelength of around 1000 nm, we developed a glass phosphor in which Yb^3+^ and Nd^3+^ were doped to Bi_2_O_3_-B_2_O_3_-based glasses. The glass phosphor was synthesized by the melt-quenching method. Powders of Yb_2_O_3_, Nd_2_O_3_, Bi_2_O_3_, and B_2_O_3_ were mixed. The mixed powders were melted at 1250 °C in an Al_2_O_3_ crucible in an electric furnace. After 10 min, the molten liquid was poured into stainless steel-molded plates kept at room temperature to enable the formation of the glass. The glass phosphor could be excited by light with a wavelength of 808 nm. Figure 1 shows the photoluminescence spectrum of the glass phosphor.

As light output is influenced by the structure of glass phosphor, we used the glass polishing method to create three shapes: rectangular (size: 1.8 mm × 1 mm × 3.5 mm), square pyramid array in two faces (size: 1.8 mm × 1 mm × 3.5 mm mm, pyramid size: 0.5 mm × 0.5 mm × 0.3 mm), and square pyramid array in four faces (size: 1.8 mm × 1 mm × 3.5 mm, pyramid size: 0.23 mm × 0.23 mm × 0.2 mm); we then compared the light outputs of the three shapes. The phosphor glass was excited using a semiconductor laser diode (SDL-808-LM-3000MFL; Shanghai Dream Lasers Technology Co., Ltd., Shanghai, China) with a wavelength of 808 nm, and the fluorescence intensity was measured using a power meter (PM100A power meter, S310C thermal power head, Thorlabs, Inc., Newton, NJ, USA) through a notch filter. The glass phosphor was irradiated with 1000 mW from a distance of 5 cm. Figure 2 shows the fluorescence intensity output of each glass phosphor structure. The square-pyramid-array-in-two-faces structure had the greatest intensity output (2.1 times greater than that of the rectangular structure). This is probably because the light emitted in the glass phosphor was refracted and focused on the large bottom area of the square pyramid.

### 2.2. Development of the Fluorescent Clip with Glass Phosphor

The fluorescent clip developed in this study consists of a commercially available hemostatic clip (HX-610-090L, Olympus Medical Systems Corp., Tokyo, Japan) as the main support and the glass phosphor with the square pyramid array in two faces structure at the tip. The glass phosphor was attached to the tip of the hemostatic clip with an adhesion bond (Figure 3).

### 2.3. Development of the Laparoscopic Fluorescent Detection System

To detect the NIR light emitted from the fluorescent chip, we developed a laparoscopic fluorescent detection system composed of a medical rigid scope (WA53000A, Olympus Medical Systems Corp., Tokyo, Japan), two notch filters, and a NIR CCD camera with a CCTV lens and camera controller (camera: C10639-80; camera controller: C2741-62; Hamamatsu Photonics K.K., Hamamatsu, Japan). The camera unit is connected to a monitor and personal computer, which makes it possible to obtain the images in real-time and record pictures and videos. A semiconductor laser diode (SDL-808-LM-3000MFL; Shanghai Dream Lasers Technology Co., Ltd., Shanghai, China) with a wavelength of 808 nm was used to excite the fluorescent clip. Figure 4 shows an overview of the laparoscopic fluorescent detection system.

### 2.4. Experimental Procedures

#### 2.4.1. Ex Vivo Detection of the Fluorescent Clips in Pig Stomach

To evaluate the performance of the fluorescent clip and the laparoscopic fluorescent detection system, we carried out an ex vivo experiment using a commercially available pig stomach. To compare the efficacy of the glass phosphor structure, a fluorescent clip with the rectangular glass phosphor structure and one with the square-pyramid-array-in-two-faces structure were used. Each fluorescent clip was fixed on the gastric mucosa and covered with stomach wall. The rigid scope (camera unit) and the excitation light source were fixed at a distance of 5 cm from the outside of the stomach. The excitation light was irradiated at 500, 1000, and 1500 mW of output power. The experiment was done using pig stomach sections with wall thicknesses of 3 and 10 mm. We carried out this kind of experiment using pig stomach in triplicate, and similar results were obtained.

#### 2.4.2. Ex Vivo Detection of the Fluorescent Clips in Human Stomach

To further evaluate the performance of the fluorescent clip and the laparoscopic fluorescent detection system, we used a freshly resected stomach (thickness: 10 mm) from a patient with gastric cancer who underwent a total gastrectomy. The study protocol was approved by the institutional ethical review board of Aichi Cancer Center (approval number: H25-3-29; approval date: 2013.6.14) and written informed consent was obtained from the patient prior to the sample collection. A fluorescent clip (square-pyramid-array-in-two-faces, glass phosphor structure) was fixed on the gastric mucosa and covered with stomach wall. The rigid scope (camera unit) and the excitation light source were fixed at a distance of 6 cm from the stomach surface. The excitation light was irradiated at an output level of 1500 mW. The detection of fluorescence was done from the proximal and distal side of the stomach wall where the clip was fixed.

#### 2.4.3. Evaluation of the Cytotoxicity of Glass Phosphor

The toxic side effect of the glass phosphor on cells was examined using normal human mesothelial cells. In this study, to examine the toxicity of this glass phosphor to the normal human tissue, we used immortalized, normal human mesothelial cells (HOMC) with stable growth potential established in our laboratory rather than cancer cells [9]. The cells (1 × 10^5^ cells/dish) were seeded on a culture dish (φ3.5 cm) and cultured in Dulbecco’s modification of Eagle medium supplemented with 10% fetal bovine serum and antibiotics at 37 °C in a 5% CO_2_ atmosphere. After 24 h, glass phosphor weights of 0, 0.5, 1.0, and 1.5 g were put into each dish and cultured for another 72 h. The cell numbers were counted with a hemocytometer in triplicate.

## 3. Results

### 3.1. Ex Vivo Detection of the Fluorescent Clips in Pig Stomach

Pig stomach was first used to evaluate the performance of the fluorescent clip and the laparoscopic fluorescent detection system. Figure 5 shows that the fluorescence intensity of the fluorescent clip increased depending on the output power of the excitation light. The glass phosphor with the square-pyramid-array-in-two-faces structure obtained a greater fluorescence output than the rectangular structure in each experimental condition. The structural effect was demonstrable when the clip was covered with a section of stomach with thicknesses of 3 and 10 mm. Thus, we decided to use the square-pyramid-array-in-two-faces, glass phosphor structure in the following investigations.

### 3.2. Ex Vivo Detection of the Fluorescent Clips in Human Stomach

Freshly resected human stomach with a thickness of 10 mm, which contained a significant amount of residual blood, was used to further evaluate the fluorescent clip and the laparoscopic fluorescent detection system in a more similar condition to the clinical setting. Figure 6b shows the detection of the fluorescent clip fixed in the stomach from the proximal and distal side of the stomach wall. The control images were obtained using a fluorescent clip fixed in the gastric mucosa without covering by the stomach. The fluorescence intensity detected from the proximal side of the stomach wall was greater than the fluorescence detected from the distal side of the stomach wall. However, sufficient fluorescence was still observed even from the distal side of the stomach wall Figure 6a, suggesting the possibility that the fluorescent clip would be effective in a clinical setting.

### 3.3. Cytotoxicity of the Glass Phosphor

We examined the cytotoxicity of the glass phosphor in culture using normal human mesothelial cells. The results showed that glass phosphor exhibited no significant changes in the cell morphology and the numbers of proliferating cells (Figure 7), suggesting that placing the clip containing the glass phosphor in the stomach for several hours has no toxic side effects on the normal human tissue.

## 4. Discussion

During laparoscopic surgery, the conventional marking method of using an endoscopic injection of charcoal ink around the tumor tissue makes it considerably difficult to identify the exact location of gastric tumors intraperitoneally because of the diffusion of the ink. This marking method generates a risk of over-resection with unnecessarily wide margins containing tumor-free tissue. To overcome this problem, two combination methods have recently been reported to precisely determine the resection margin. One method is intraoperative endoscopy in combination with laparoscopic surgery [10], and the other is preoperative endoscopic clipping in combination with intraoperative radiography [11]. However, these two methods require an additional diagnostic apparatus and manpower. In contrast, the new fluorescent clip with glass phosphor proposed in the present study emits NIR light (around 1000 nm), which passes through the gastric wall and can be detected by fluorescent laparoscopy alone. This new method does not need further equipment and/or manpower but has the potential to accurately identify gastric tumors during laparoscopic surgery. To the best of our knowledge, this is the first report of a combined NIR light-emitting clip and laparoscopic detection system with the ability to visualize a precise tumor location during laparoscopic surgery of gastric cancer.

Our fluorescent clip with glass phosphor has the following two unique characteristics: (1) The wavelength emitted by the glass phosphor is known to be dependent on the composition of the doping elements. We previously fabricated Yb^3+^ and Nd^3+^ co-doped Bi_2_O_3_-B_2_O-based glass as a new NIR fluorescent light source [12]. As the luminescence of Yb^3+^ and Nd^3+^ is located in the wavelength region around 1000 nm after excitation by light (808 nm) [13], we applied this glass phosphor in the clip as a fluorescence light source that can pass through stomach wall and enable the identification of the location of the tumor from outside the stomach. (2) The fluorescence intensity of emitted NIR light from glass phosphor was substantially influenced by the three-dimensional geometrical structure of the glass phosphor. Kasugai et al. previously reported that light extraction efficiency was dependent on the surface structure of light-emitting diodes, such as the moth-eye structure [14]. Chan et al. also recently reported that the moth-eye structure inspired anti-reflective surfaces for improved IR optical systems [15]. Therefore, we examined the influence of the structure of glass phosphor and found that among various structures, the square pyramid array in two or four faces exhibited a significantly greater light intensity output than the simple rectangular structure. In this study, therefore, we used the clip with Yb^3+^ and Nd^3+^ co-doped glass phosphor in the aforementioned pyramid array structure.

The important finding of the present study was the acquisition of preclinical proof of the utility of the proposed fluorescent clip and laparoscopic detection system using two ex vivo experimental models, including resected pig and human stomach. The fluorescence intensity emitted from the clip through the stomach wall increased depending on the excitation light power (500–1500 mW) in both experiments. In the ex vivo experiment using a resected human stomach, the thickness of the stomach was 10 mm on average, which was thicker than the pig stomach (3–10 mm). In addition, the freshly resected human stomach tissue still contained a significant amount of blood unlike the commercially available pig stomach. Even in this severe condition, a strong fluorescence intensity was successfully detected from outside of the human stomach when the excitation light and rigid scope were close to the stomach wall and fixed at distances of about 6 cm from the outside of the stomach, suggesting the possibility that this combined NIR light-emitting clip and laparoscopic detection system is applicable to laparoscopic surgery in clinical settings.

In conclusion, there are still some limitations to clearly detecting NIR fluorescence from a greater distance (6–8 cm) outside of the stomach. Further improvement of the sensitivity of the laparoscopic NIR light detection system, especially by increasing the transmission efficiency of the NIR light through the rigid laparoscope, is needed. Nevertheless, the present findings strongly suggest that the proposed clip with glass phosphor and the laparoscopic detection system could be useful and practical diagnostic tools for navigation during laparoscopic gastric surgery in the clinical setting in the near future.

## Figures and Tables

**Figure 1 micromachines-10-00081-f001:**
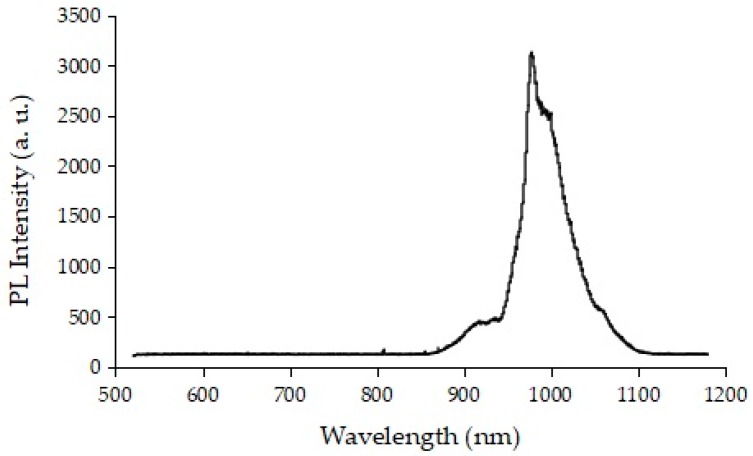
Photoluminescence spectrum of Yb^3+^ and Nd^3+^ co-doped Bi_2_O_3_-B_2_O_3_ glass phosphor. Light with a wavelength of 808 nm is suitable for excitation.

**Figure 2 micromachines-10-00081-f002:**
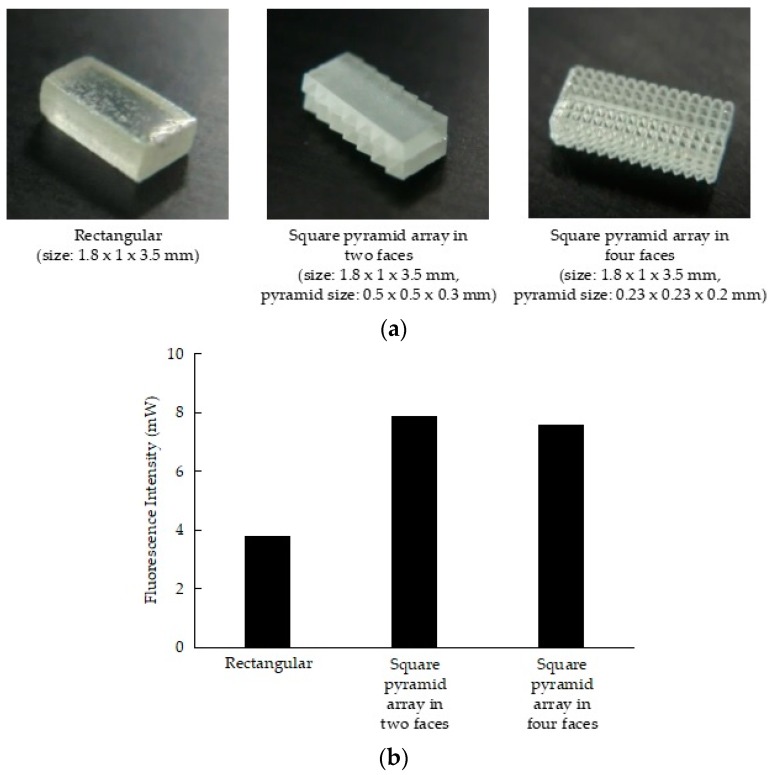
Morphological structures of the three glass phosphor shapes created and their respective fluorescence intensity outputs. (**a**): Rectangular, square-pyramid-array-in-two-faces and square-pyramid-array-in-four-faces structures were made using the glass polishing method. (**b**): The fluorescence intensity of each glass phosphor structure was measured at a distance of 5 cm and an irradiation intensity of 1000 mW using a semiconductor laser diode (808 nm). The square pyramid array with two faces showed the greatest intensity output.

**Figure 3 micromachines-10-00081-f003:**
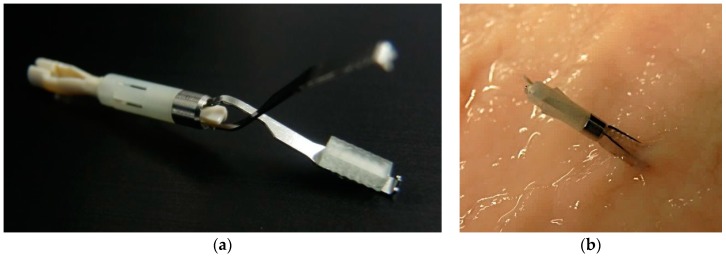
Photographs of (**a**) the fluorescent clip with glass phosphor at the tip and (**b**) a glass phosphor-attached clip fixed to the stomach mucosa.

**Figure 4 micromachines-10-00081-f004:**
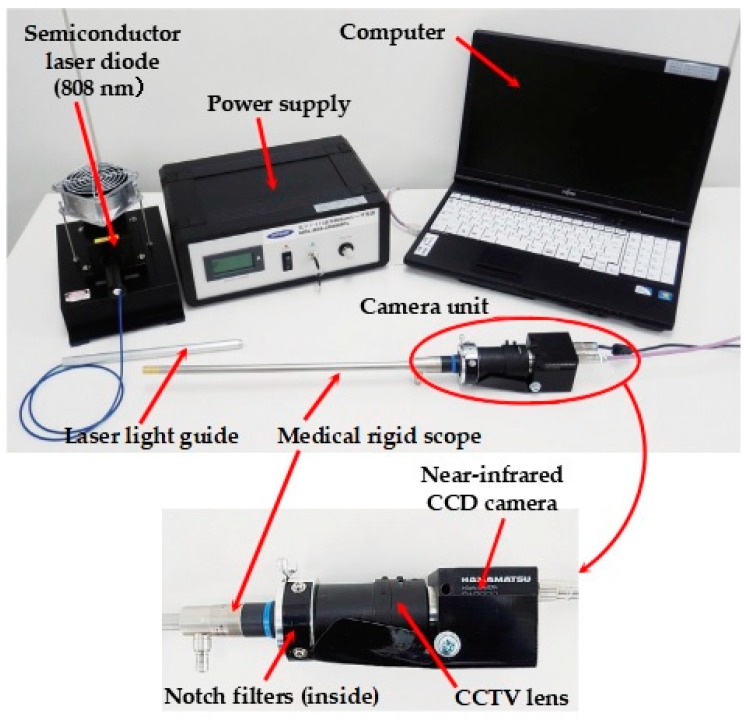
The laparoscopic fluorescent detection system composed of a medical rigid scope with a near-infrared CCD camera, a semiconductor laser diode (808 nm) as an excitation light source, and a computer with monitor.

**Figure 5 micromachines-10-00081-f005:**
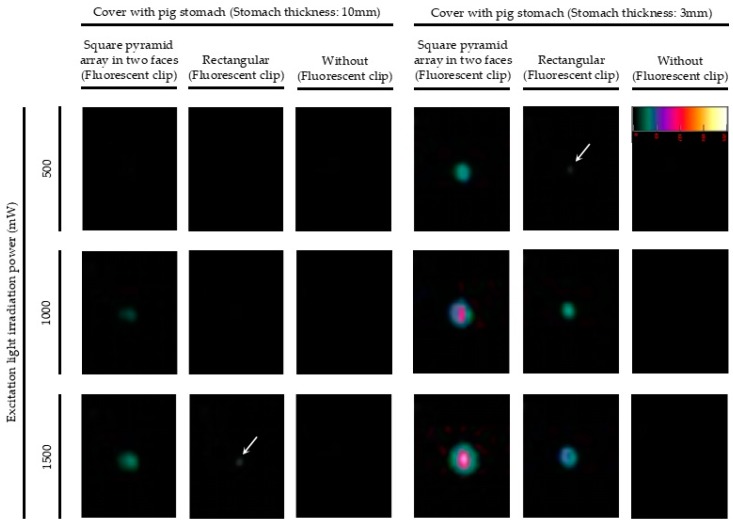
Detection of the fluorescent clips fixed in a commercially available pig stomach sample. The fluorescent image of the clip with the square-pyramid-array-in-two-faces, glass phosphor structure was successfully obtained with good fluorescence output.

**Figure 6 micromachines-10-00081-f006:**
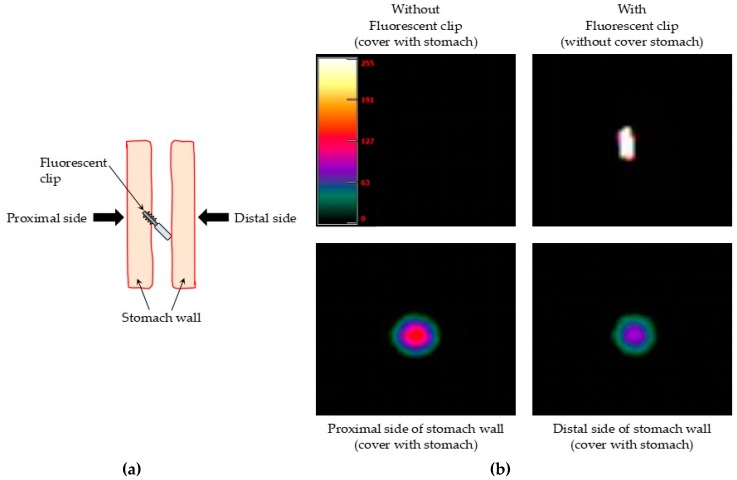
Ex vivo detection of the near-infrared fluorescent clips fixed in human stomach. (**a**) Schematic representation of the detection method of the fluorescent clips fixed in the human stomach. (**b**) A stronger fluorescent image of the clip from the proximal side of the stomach wall than from the distal side was observed.

**Figure 7 micromachines-10-00081-f007:**
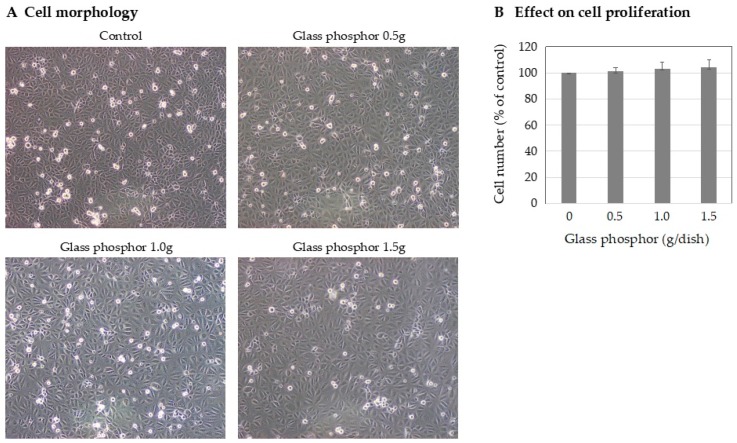
Effects of glass phosphor on the normal human mesothelial cells in culture. The cells were cultured in dishes containing glass phosphor (weights: 0.5, 1.0, and 1.5 g) for 72 h. (**A**) No significant change in the cell morphology was observed. (**B**) There was statistically no significant difference (NS) in the cell proliferation (by *t*-test). Bar = standard deviation.

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
