# Peer review of "Development of a New Laparoscopic Detection System for Gastric Cancer Using Near-Infrared Light-Emitting Clips with Glass Phosphor"

_micromachines, 2019, doi:10.3390/mi10020081_

Round 1

Reviewer 1 Report

The manuscript by Inada et al seems an extension of their previously published paper in Journal of Physics (2015) with almost the same title. This version describes the use of the NIRF clip with a prototype NIRF-II camera system in a pre-clinical setting. Although the new data are certainly relevant, the message is relatively thin and meant for a relatively small audience.

Minor points:

The part of the introduction defining the clinical problem (from line 53) needs references.

The M&M section contains data (e.g. figure 1) which is probably because they were previously published. The source and choice of the cell line should be given, especially because these immortalized cell lines are used for toxicity studies.

The legend of figure 4 needs to indicate a computer/PC.

The problem discription in the results section (line 224) should be integrated in the introduction.

Figure 9 needs a titel and figure b is in the wrong format for this kind of experiments.

A large part of the discussion seems repetition of introduction and results section. This should be shortened. The data are not over-interpreted but could need a more clinical focus, e.g. which other tumors/applications could be served.

Some of the references like 1, 2, 13 are out-dated or akward.

Author Response

Thank you for  your comments. Please see our responses in attachment.

Reviewer 2 Report

1. The innovation of the work should be indicated more clearly.

2. Recent references concerning the B2O3-Bi2O3 containing  gasless have show their very promising optical and radiation properties [M.I. Sayyed,, G. Lakshminarayana, M.G. Dong, M. Çelikbilek Ersundu, A.E. Ersundu  Investigation on gamma and neutron radiation shielding parameters for BaO/SrO‒Bi2O3‒B2O3 glasses. Radiation Physics and Chemistry 145 (2018) 26–33]

3. The accuracy of the data reproduction should be added.

4. The Conclusions should be more sound.

Author Response

(The authors gave the same response as above.)

Round 2

Reviewer 1 Report

accept

Reviewer 2 Report

Accept as it is.